# Amphiphilic Molecular Brushes with Regular Polydimethylsiloxane Backbone and Poly-2-isopropyl-2-oxazoline Side Chains. 3. Influence of Grafting Density on Behavior in Organic and Aqueous Solutions

**DOI:** 10.3390/polym14235118

**Published:** 2022-11-24

**Authors:** Serafim Rodchenko, Mikhail Kurlykin, Andrey Tenkovtsev, Sergey Milenin, Maria Sokolova, Alexander Yakimansky, Alexander Filippov

**Affiliations:** 1Institute of Macromolecular Compounds of the Russian Academy of Sciences, Bolshoy pr., 31, 199004 Saint Petersburg, Russia; 2Research Laboratory of New Silicone Materials and Technologies, Tula State Lev Tolstoy Pedagogical University, Lenin Avenue, 125, 300026 Tula, Russia; 3Enikolopov Institute of Synthetic Polymeric Materials of the Russian Academy of Sciences, Profsoyuznaya, 70, 117393 Moscow, Russia

**Keywords:** poly-2-isopropyl-2-oxazoline, polydimethylsiloxane, molecular brushes, thermosensitivity, light scattering, turbidimetry, compaction, aggregation, LCST

## Abstract

Regular and irregular molecular brushes with polydimethylsiloxane backbone and poly-2-isopropyl-2-oxazoline side chains have been synthesized. Prepared samples differed strongly in the side chain grafting density, namely, in the ratio of the lengths of spacer between the grafting points and the side chains. The hydrodynamic properties and molecular conformation of the synthesized grafted copolymers and their behavior in aqueous solutions on heating were studied by the methods of molecular hydrodynamics and optics. It was found that the regularity and the grafting density do not affect the molecular shape of the studied samples of molecular brushes in the selective solvent. On the contrary, the grafting density is one of the most important factors determining the thermoresponsivity of grafted copolymers. It was shown that in analyzing self-organization and LCST values in aqueous solutions of poly-2-isopropyl-2-oxazolines with complex architecture, many factors should be considered. First is the molar fraction of the hydrophobic fragment and the intramolecular density. It was found that molar mass is not a factor that greatly affects the phase transition temperature of poly-2-isopropyl-2-oxazolines solutions at a passage from one molecular architecture to another.

## 1. Introduction

At present, stimuli-responsive polymers attract ever-increasing attention due to the wide possibilities of their practical application. They are used in biotechnology, bioengineering, and medicine, in particular, for targeted drug delivery [1,2,3,4,5]. A key feature of stimuli-responsive polymers is the nonlinear response to external stimuli, i.e., a strong change in characteristics under very slight external influences.

In the case of water-soluble thermosensitive polymers, the response is due to a change in the hydrophilic-hydrophobic balance with temperature variation. This leads to self-organization at the molecular and supramolecular levels. For the vast majority of thermoresponsive polymers, the described processes are reversible, which makes it possible to control both the formation (or destruction) of supramolecular structures and the association of such polymers with low molecular weight compounds.

Thermoresponsive polymers are divided into polymers with upper and lower critical solution temperatures (UCST and LCST, respectively) according to the nature of the response [6,7]. For medical applications, thermosensitive polymers with LCST, which is close to the human body temperature, are of the most interest.

The thermoresponsive behavior of polymers is conveniently regulated by copolymerization using comonomers that differ in hydrophobicity. The inclusion of a more hydrophilic comonomer increases phase separation temperature and LCST, while a more hydrophobic comonomer leads to a decrease in these temperatures [8,9]. Accordingly, variation of the comonomer content allows controlling the phase separation temperature. For example, the LCST of aqueous solutions of gradient amphiphilic copolymers of 2-ethyl- and 2-isopropyl-2-oxazoline increases linearly with increasing mole fraction of 2-ethyl-2-oxazoline [10]. For copolymers of 2-isopropyl-2-oxazoline with other 2-alkyl-2-oxazolines (PAlOx) (*n*-propyl-, *n*-butyl-, and *n*-nonyl-2-oxazoline), it was found that a wide temperature range of LCST (from 9 to 75 °C) can be realized [11]. On the other hand, the literature data concerning these temperature changes for a given chemical structure and composition in the series block copolymer-gradient copolymer-statistical copolymer are contradictory [10,12].

In aqueous solutions of thermoresponsive amphiphilic block copolymers, the formation of aggregates caused by the interaction of hydrophobic components is often observed already at low temperatures. Moreover, several types of particles can be observed simultaneously in copolymer solutions at room temperature; there are individual macromolecules, micelle-like aggregates, and large loose supramolecular structures [13,14,15,16,17].

Of particular interest are thermosensitive polymers with complex architecture and physico-chemical properties that depend on the architecture parameters, such as the rigidity of the main chain, the size and grafting density of the side chain, etc. Usually, macromolecules with complex architecture combine components of different chemical natures, opening up new possibilities for the precision regulation of properties. For example, in the case of polymer brushes, the main and side chains can be polymers of different chemical classes [18,19,20]. Accordingly, numerous polymers with complex architecture are copolymers. Note that amphiphilic copolymers with complex architecture appear to be more interesting in comparison with linear copolymers since the self-organization of their molecules in solutions is determined not only by the chemical structure but also by the architecture.

Among polymers with complex architecture, molecular brushes are of particular interest. The term “molecular” means that each brush is an individual macromolecule. The main feature of the structure of cylindrical molecular brushes is the presence of two structural elements, namely, the backbone and the side chains, which are covalently attached to each other [21,22,23,24]. The behavior of cylindrical brushes strongly depends on such parameters as the ratio of the lengths of the backbone and side chains, the linear grafting density, the chemical nature of the components and their content, etc. For example, if grafting density is high, the macromolecule takes on an elongated conformation due to the steric repulsions of side chains. On the contrary, in the case of amphiphilic loose brushes, the backbone interaction can lead to aggregation. Molecular brushes are promising for use in the field of intramolecular nanoengineering, for example, as nanocontainers for inorganic nanoparticles or nanofibers, as carrier polymers for controlled drug delivery [25,26,27].

Significant progress has been made in the preparation and study of amphiphilic molecular brushes [28,29,30]. Their behavior is determined not only by structural parameters but also by the thermodynamic quality of the solvent with respect to the main and side chains. By tuning the thermodynamic quality of the solvent, both the conformation and the aggregation of the macromolecule can be controlled. Note that most thermoresponsive molecular brushes are amphiphilic copolymers [31,32,33,34,35,36,37,38,39,40,41], where the thermosensitivity depends on the architecture parameters. Consequently, the self-organization and the phase separation temperature *T*_pt_ in molecular brush solutions are influenced strongly by the grafting density and the molar mass of the backbone and side chains.

For molecular brushes with PAlOx side chains and polymethacrylate backbone, the effect of composition on the copolymer’s ability to form unimers or aggregates in water, methanol, and ethanol was found [35]. For copolymers with poly-2-isopropyl-2-oxazoline (PiPrOx) side chains grafted onto the polyacrylamide, an increase in intramolecular density with increasing solution temperature was reported [36]. Grafted copolymers with polymethacrylate backbone and poly-2-ethyl-2-oxazoline side chains demonstrated the simultaneous coexistence of aggregates and molecules in solutions at low temperatures [37]. In addition, for thermoresponsive molecular brushes with PAlOx side chains, it was also found that *T*_pt_ decreases with backbone lengthening [38].

A study of the behavior of grafted thermoresponsive copolymers with an aromatic polyester backbone and poly-2-ethyl-2-oxazoline side chains, differing in the grafting density *z*, showed an increase in the *z* value prevents the interaction between the hydrophobic polyester chains [39]. For the grafted copolymer with high *z*, the molecular compaction prevailed in aqueous solutions at *T* < *T*_pt_. Macromolecule aggregation was observed throughout the temperature range studied for the loose brush. The interaction of the backbone of the different molecules can explain it. The phase separation temperatures increased with dilution for both discussed copolymers and reduced with *z* decreasing due to an increase in macromolecular hydrophobicity [32,39].

Despite considerable progress in the approaches to the synthesis of molecular brushes, systematic studies of their physico-chemical properties are just beginning, and many problems remain unresolved. The experimental material accumulated so far does not allow us to draw strong quantitative conclusions concerning the behavior of stimuli-sensitive cylindrical molecular brushes.

Our previous works described the synthesis, characterization, and conformational behavior of amphiphilic PDMSk-*graft*-PiPrOx molecular brushes with polydimethylsiloxane (PDMS) backbone and poly-2-isopropyl-2-oxazoline (PiPrOx) side chains, and the effect of PiPrOx side chain length on self-organization and aggregation in aqueous solutions was also investigated [42,43]. It was shown that the LCST for PDMSk-*graft*-PiPrOx with long side chains is higher than the LCST for samples with short PiPrOx chains, despite the fact that the *M* of the latter is almost two times lower. Consequently, the molecular mass is not a crucial factor in the thermoresponsibility of amphiphilic molecular brushes. This work aimed to find the influence of grafting density on the hydrodynamic and conformational characteristics and self-organization in water solutions of PDMSk-*graft*-PiPrOx brushes. For these two samples of PDMSk-*graft*-PiPrOx differing strongly in the *z* value were synthesized and investigated. The results are compared with data for previously investigated samples of PDMSk-*graft*-PiPrOx [42,43].

## 2. Materials and Methods

### 2.1. Characteristics of Investigated Samples

The structure of the monomer units of the synthesized and studied grafted copolymers is shown in Figure 1. The main structural difference between the investigated samples of PDMS3-*graft*-PiPrOx and PDMS10-*graft*-PiPrOx is the difference in the structure of the PDMSk macroinitiators. In molecules of PDMS3, which was used to prepare PDMS3-*graft*-PiPrOx, the functional undecyl tosylate groups are in every third –SiO- unit, i.e., *k* = 3 (Figure 2). In the case of PDMS10 and PDMS10-*graft*-PiPrOx, the segment of the polydimethylsiloxane chain between neighboring functional groups is much longer. The *k* value is 10 on average (Figure 2). In addition, PDMS3 has a regular structure (*k* = const = 3), while PDMS10 is an irregular polymer; the length of the mentioned segment varies somewhat along the chain (*k* ≠ const).

### 2.2. Synthesis of Multicenter Macroinitiator PDMS3

Then, 1 μL of Carstedt catalyst (platinum (0)-1,3-divinyl-1,1,3,3-tetramethyldisiloxane complex solution in xylene, Pt ≈ 2%) was added to a solution of 0.31 g (1.1 mmol) of PDMS and 0.3547 (1.1 mmol) of 10-undecene-1-tosylate in 2 mL of toluene. The reaction mixture was stirred at room temperature overnight. After finishing the reaction, the toluene was removed under reduced pressure using a vacuum pump to concentrate the product. The product was used without further purification. The yield was quantitative. ^1^H NMR (300 MHz, CDCl_3_, δ, ppm): 7.77–7.34 (m, -SC_6_H_4_CH_3_), 4.00 (m, O-CH_2_CH_2_), 2.44 (m, -SC_6_H_4_CH_3_), 1.62 (m, O-CH_2_CH_2_), 1.20 (m, SiCH_2_(CH_2_)_8_-), 0.4 (m, SiCH_2_(CH_2_)_8_-), 0.05 (m, Si-CH_3_).

### 2.3. Synthesis of Nonregular Macroinitiator PDMS10

After that, 1 μL of Carstedt catalyst (platinum (0)-1,3-divinyl-1,1,3,3-tetramethyldisiloxane complex solution in xylene, Pt ≈ 2%) was added to a solution of 1.3 g (1.6 mmol) of PDMS and 0.52 g (1.6 mmol) of 10-undecene-1-tosylate in 4 mL of toluene. The reaction mixture was stirred at room temperature overnight. After finishing the reaction, the toluene was removed under reduced pressure using a vacuum pump to concentrate the product. The product was used without further purification. The yield was quantitative. ^1^H NMR (300 MHz, CDCl_3_, δ, ppm): 7.77–7.34 (m, -SC_6_H_4_CH_3_), 4.00 (m, O-CH_2_CH_2_), 2.44 (m, -SC_6_H_4_CH_3_), 1.62 (m, O-CH_2_CH_2_), 1.20 (m, SiCH_2_(CH_2_)_8_-), 0.4 (m, SiCH_2_(CH_2_)_8_-), 0.05 (m, Si-CH_3_).

### 2.4. Polymer Brushes Synthesis

An ampoule with the given amount of initiator (about 200 mg), the rated amount of 2-isopropyl-2- oxazoline (initiator: monomer ratio of 1: 30 by functional groups), and 3 mL dichloroethane were frozen to −196 °C; the air was removed under vacuum (0.1 mm Hg); then the mixture was defrosted under argon. The cycle was repeated three times; after that, the ampoule was sealed and heated at 100 °C for 120 h. Then the ampoule was opened, a 50% aqueous ethanol (1 mL) was added, and the ampoule was kept at room temperature for 24 h. The solvent was eliminated under vacuum (0.1 mm Hg) at room temperature. The product was dissolved in water, dialyzed against water for 24 h, and subjected to lyophilic drying.

^1^H NMR (CDCl_3_): δ, ppm): 0.05 (s, -Si(CH_3_)_2_OSi(CH_3_)_2_) 0.08 (t, SiCH_2_(CH_2_)_8_CH_2_), 1.13 (br, CH(CH_3_)_2_), 1.20–1.38 (m, SiCH_2_(CH_2_)_8_CH_2_), 2.64 and 2.89 (br, rotamers CH(CH_3_)_2_), 3.43 (br, NCH_2_CH_2_).

The NMR spectra were measured on the Bruker AC400 (400 MHz) (Bruker, Billerica, MA, USA) spectrometer using chloroform solutions. For the purification of the synthesized star polymers Zellu Trans dialysis tubes (Scienova GmbH, Germany. Material: cellulose, regenerated) with MWCO 12,000–14,000 g∙mol^−1^ were used. Chromatographic analysis was performed on the Shimadzu LC-20AD chromatograph (Shimadzu Corporation, Nishinokyo Kuwabara, Japan) equipped with the refractive detector. Linear samples were studied using the PSS SDV column (5 μm, 8 × 300 mm) (Polymer Standards Service GmbH, Mainz, Germany) in THF solutions at 40 °C. The calibration curve was obtained using polystyrene standards). Grafted copolymers were studied using the Waters Styragel HT4 column (10 μm, 7.8 × 300 mm) (Waters Chromatography Ireland Ltd, Dublin, Ireland) in dimethylformamide solutions at 60 °C without a calibration curve.

### 2.5. Investigation of Molecular Dispersed Solution of PDMSk and PDMSk-graft-PiPrOx

The methods of static (SLS) and dynamic (DLS) light scattering and viscosimetry were used to determine the molar mass (MM) and hydrodynamic characteristics of the investigated polymers. Chloroform (density ρ_0_ = 1.489 g∙cm^−3^, dynamic viscosity η_0_ = 0.570 cP, refractive index *n*_0_ = 1.4455) for PDMS3 and PDMS10 and 2-nitropropane (ρ_0_ = 0.9884 g∙cm^−3^, *n*_0_ = 1.3944, η_0_ = 0.790 cP) for PDMSk-*graft*-PiPrOx were chosen as solvents because no associative phenomena were observed in them. The experiment temperature was equal to 21.0 °C. The polymer solutions and the solvents were filtered before measurements through Chromafil syringe filters (“Macherey-Nagel”, Düren, Germany) with a polytetrafluoroethylene membrane with pores of 0.45 or 0.2 μm.

Light scattering experiments were performed on the Photocor Complex (Photocor Instruments Inc., Moscow, Russia). The light source was a Photocor-DL diode laser (wavelength λ = 659.1 nm). The correlation function of light scattering was obtained using a Photocor-PC2 correlator with the number of 288 channels and processed using DynaLS soft (ver. 8.2.3, SoftScientific, Tirat Carmel, Israel).

For all the studied solutions, a unimodal distribution of the scattered light scattering intensity *I* over the hydrodynamic radii *R*_h_ was recorded (Figure 3). The *R*_h_ value did not depend on the solution concentration *c*, and the concentration-averaged *R* value was taken as the hydrodynamic radius of *R*_h-D_ macromolecules.

There was no asymmetry of scattered light in the range of scattering angles from 45° to 135°, so the weight average molar mass *M*_w_ and second virial coefficient *A*_2_ were found by the Debye method:*Hc*/*I*_90_ = 1/*M_w_P*(90°) + 2*A*_2_*c*,(1)
where *I*_90_ is the excess light scattering intensity at an angle of 90°, *P*(90°) is the Debye scattering factor (for an angle of 90° it is 1), and *H* is the optical constant:*H* = *4*π^2^*n*_0_^2^(*dn/dc*)^2^/*N_A_*λ^4^,(2)
where *dn*/*dc* is the refractive index increment and *N*_A_ is the Avogadro number. Figure 4 shows the Debye dependencies for the polymers studied. Positive values of *A*_2_ indicate good thermodynamic quality of the selected solvents for the studied polymers.

The *dn*/*dc* values were measured using an RA-620 refractometer (KEM, Japan) with λ = 589.3 nm. The refractive indices of the solvent *n* and solutions *n*_0_ were measured with an accuracy of 0.00001. The refractive index increment was determined from the slope of the dependences of ∆*n* = *n* − *n*_0_ on concentration *c* (Figure 5).

The intrinsic viscosity [η] is related to the rotational friction coefficient of the particle and is very sensitive to changes in the shape, size, and density of dissolved objects. Accordingly, the analysis of the [η] values and their molar-mass dependence allows us to make conclusions about the molecular conformation. Ostwald glass viscometers and a microVISC viscometer (Rheosense, USA) were used to measure the characteristic viscosity. The flow time of the solvent *t*_0_ was 48.8 s and 100.1 s for chloroform and 2-nitropropane, respectively. The experimental data are well described by the Huggins equation:η_sp_/*c* = [η] + *k*′[η]^2^*c*,(3)

The Huggins constant *k*′ characterizes the thermodynamic interaction between the solvent and polymer, as well as the hydrodynamic behavior of the solution [44,45,46].

### 2.6. Investigation of Aqueous Solutions of PDMSk-graft-PiPrOx

Self-organization in aqueous solutions of PDMSk-*graft*-PiPrOx was studied by light scattering and turbidimetry using the Photocor Complex described above, which is equipped with the Photocor-PD detection device for measuring the transmitted light intensity. The solutions were filtered into dust-free vials using Millipore filters with hydrophilic H-PTFE membrane and pore size of 0.45 µm (Merck KGaA, Darmstadt, Germany). The studies were carried out in the temperature range *T* from 15 to 75 °C. The *T* value was changed discretely with a step from 0.5 to 5 °C with an accuracy of 0.1 °C.

After the given temperature was established, the dependences of the light scattering intensity *I* and optical transmission *I** on time *t* were recorded at a scattering angle of 90°. At each temperature, it is assumed that *t* = 0 is the moment when the sample reaches the required temperature. *I* and *I** reach the constant values at time *t*_eq_. For the studied PDMSk-*graft*-PiPrOx, the *t*_eq_ values reached 15,000 s. Under “equilibrium” conditions (*I* and *I** = const), the hydrodynamic radii *R*_h_ of the scattering objects and the contribution *S*_i_ of each type of scattering particle to the total light scattering intensity were determined. Under these conditions, the angular dependences of the *I*, *R*_h_, and *S*_i_ values in the range of the angles of light scattering from 45° to 135° were investigated in order to prove the diffusive nature of the modes, as well as to obtain extrapolated values of *R*_h_ and *S*_i_, which were used in further discussion.

### 2.7. Microscopic Investigation

The atomic force microscopy (AFM) measurements were conducted with an SPM-9700HT (Shimadzu, Kyoto, Japan) scanning probe microscope. Collected data were analyzed with the SPM software v.4.76.1 (Shimadzu, Kyoto, Japan). The measurements were performed in the dynamic regime under atmospheric conditions using NSG10-SS silicon cantilevers (TipsNano, Moscow, Russia) with a 2 nm tip radius.

## 3. Results and Discussion

### 3.1. Synthesis of Macroinitiator PDMS3

The present work used a hydride-containing polydimethylsiloxane polymer of the regular structure obtained in our previous work to synthesize a hybrid polymer brush [42]. In turn, the polydimethylsiloxane polymer was obtained by the interaction of the sodium salt of 1,5-disodiumoxyhexamethyltrisiloxane [47] with dimethyldichlorosilane, followed by fractionation of the resulting product. The fraction with *M* = 45,000 g∙mol^−1^ and the dispersity *Ð* = 1.3 (obtained by GPC) was used to obtain a macroinitiator based on the synthesized polydimethylsiloxane matrix. The macroinitiator was obtained by hydrosilylation of 10-undecene-1-tosylate in a toluene medium in the presence of Karstedt’s catalyst. The average molar mass for the obtained tosylate derivative *M* = 77,000 g∙mol^−1^ (obtained by DLS).

### 3.2. Synthesis of Macroinitiator PDMS10

Additionally, a macroinitiator based on PDMS of the irregular structure was synthesized in this work using a similar technique by hydrosilylation of 10-undecene-1-tosylate with appropriate PDMS with hydridesilyl groups distributed along the polymer chain. In addition to the irregularity of the structure, the number of functional fragments in the chain correlated with dimethylsiloxane fragments was approximately 1:10, as confirmed by ^1^H NMR spectra (Figure 6). The molecular weight of the polymer was *M*_w_ = 10,300 g∙mol^−1^ (obtained by GPC).

Thus, two macroinitiators were synthesized, with 10-undecene-1-tosylate initiating groups based on PDMS of regular and irregular structure.

### 3.3. Synthesis of Molecular Brushes

The grafting procedure was modified in comparison with the typical one [42]. Namely, in the modified procedure, dichloroethane was used as the solvent instead of acetonitrile due to the poor solubility of the macroinitiator. As a result, two samples of molecular brushes were obtained using the macroinitiators described above with the ratio of functional fragments to dimethylsiloxane link as 1:3 (grafted copolymer PDMS3-*graft*-PiPrOx) and 1:10 (PDMS10-*graft*-PiPrOx), respectively. The relative grafting density of polyoxazoline fragments, namely, the fraction *z* of monomer units of the backbone containing a side chain, can be estimated from the intensity ratio of the CH_2_OTs signal at 3.72 ppm and of the CH_3_-Si signal at 0.0–0.1 ppm in the ^1^H NMR spectra of the macroinitiators and the grafted copolymer (Figure 7). This intensity ratio decreases by 70 and 60% for PDMS3-*graft*-PiPrOx and PDMS10-*graft*-PiPrOx, respectively, corresponding to the initiation of polymerization on every 1.5–2 tosylate group. However, it should be taken into account that the accuracy of determining the *z* values is not high and about 10–15%.

### 3.4. Structural Characteristics of Macroinitiators and Grafted Copolymers

For an adequate interpretation of the results of studying the self-organization and aggregation of thermoresponsive polymers in aqueous solutions, it is necessary to have complete information about their hydrodynamic, conformational, and molar mass characteristics. In turn, the analysis of the polymer conformation is impossible without knowledge of the structural parameters of the samples under study. Molecular and hydrodynamic characteristics obtained for the investigated samples are given in Table 1.

Note, for all the studied samples, the values of the hydrodynamic radius *R*_h-D_ are in good agreement with the data of microscopic studies. For example, AFM studies demonstrated that a PDMS3-*graft*-PiPrOx sample prepared from 2-nitropropane solution on a mica surface (Figure 8) consists of spherical particles with a diameter from 25 to 32 nm, which form a nearly continuous coating on the substrate surface.

The length of the macroinitiator repeating unit λ_bb_ or the distance between neighboring side functional 10-undecene-1-tosylate groups depends on the length λ_0 PDMS_ = 0.26 nm [48] of the PDMS monomeric units and the number *k* of these units in the repeating unit of PDMSk:λ_bb_ = λ_0-PDMS_·*k*(4)

As can be seen from Table 2, the λ_bb_ value for PDMS3 is three times greater than λ_bb_ for PDMS10. Macroinitiators are comb-like polymers, the side chain of which is a functional 10-undecene-1-tosylate group. The side chain length *L*_sc_ of PDMSk is equal to the sum of the lengths of the –(CH_2_)_10_– chain and the tosylate group. The *L*_sc_ values are easy to estimate, assuming that all bond angles are tetrahedral and the bond lengths are 0.14 nm. In this approximation, *L*_sc_ = 2.14 nm. For the studied PDMSk, the values of λ_bb_ and *L*_sc_ are the same in order of magnitude, and the ratio *L*_sc_/λ_bb_ is 2.7 and 0.8 nm for PDMS3 and PDMS10, respectively. Therefore, the PDMSk are comb-like polymers with rare side chains, the steric interactions of which are minimal.

An important structural characteristic of grafted copolymers is the average distance ∆*L* along the PDMS backbone between two neighboring side chain grafting points. The ∆*L* value is easy to calculate, knowing the relative grafting density *z* of the PiPrOx side chains and the length of the macroinitiator repeating unit:Δ*L* = λ_bb_/*z*(5)

*z* is the relative fraction of the monomeric units of the backbone containing grafted PiPrOx chains. The Δ*L* values for PDMS3-*graft*-PiPrOx and PDMS10-*graft*-PiPrOx differ by almost four times (Table 2).

In order to understand how strong the steric interactions of side chains are, it is necessary to compare Δ*L* with the side chain length *L*_sc_. The latter consists of the lengths of the PiPrOx chains *L*_PiPrOx_ and the spacer connecting the backbone and PiPrOx:*L*_sc =_ *L*_PiPrOx_ + *L*_sp_ = *N*_PiPrOx_·λ_0-PiPrOx_ + *L*_sp,_(6)
where λ_0-PiPrOx_ = 0.378 nm is the monomer unit length of the *L*_PiPrOx_ and *L*_sp_ = 1.44 nm is the spacer length. The degree of polymerization *N*_PiPrOx_ of PiPrOx chains can be calculated using the formula:*N*_PiPrOx_ = (*M*_gsp_ − *M*_bb_)/(*z·N*_bb_·*M*_0-PiPrOx_)_,_(7)
where *M*_gsp_ and *M*_bb_ are the MM of PDMSk-*graft*-PiPrOx grafted copolymers and their backbone, *M*_0-PiPrOx_ = 113 g·mol^−1^ is the MM of the PiPrOx monomer unit. For both graft copolymers, the *L*_sc_/∆*L* ratio is relatively small (Table 2). Therefore, the PDMSk-*graft*-PiPrOx can be considered as loose molecular brushes.

The number of grafted chains *f*_sc_ in PDMS3-*graft*-PiPrOx molecular brushes is equal to the ratio of the backbone length *L*_bb_ to the average distance ∆*L* along the PDMS chain between two neighboring grafting points of side PiPrOx chains. Table 2 shows that the number of grafted chains in PDMS3-*graft*-PiPrOx molecules is 15 times higher than *f*_sc_ for PDMS10-*graft*-PiPrOx.

An important parameter that determines the polymer behavior in aqueous solutions is the mass fraction ω of the hydrophobic part, which can be calculated using the *M* of the graft copolymer and its hydrophobic fragments:ω = *M*_phob_/*M*_gsp_(8)
where *M*_phob_ is the total MM of hydrophobic fragments in PDMSk-*graft*-PiPrOx macromolecules. For the studied copolymers, the values of ω are large and differ by 2.5 mass% (Table 2). The content of PiPrOx affects the refractive index increment *dn*/*dc* of the solutions of the studied cylindrical molecular brushes: the larger the ω, the higher the *dn*/*dc* value (Table 2). This behavior is explained by the difference in the refractive indexes of the PiPrOx and PDMS. For macroinitiators, the value of the partial specific volume $\bar{v}$ is smaller than the $\bar{v}$ for linear PDMS and close to $\bar{v}$ = 0.97 cm^3^ g^−1^ for comb-like PDMS with short side chains [49]. The grafting of side PiPrOx chains leads to a decrease in $\bar{v}$. The greater the value, the smaller the value of ω (Table 2) because the $\bar{v}$ of the linear PiPrOx is small. Its value ranges from 0.84 to 0.87 cm^3^ g^−1^ [50,51,52].

### 3.5. Hydrodynamical and Conformational Characteristics of Macroinitiators and Grafted Copolymers

As can be seen from Table 1, the intrinsic viscosity for the PDMS3 and PDMS10 is greater than the [η] value for PDMSk-*graft*-PiPrOx, despite the fact that the MM of the grafted copolymers is much greater than the MM of the macroinitiators. This fact alone indicates a higher intramolecular density of the grafted copolymers under consideration compared to their linear analogs. The high density of macromolecules of the studied brushes is confirmed by the low values of the hydrodynamic invariant *A*_0_ [44,53,54]:*A*_0_ = (η_0_*D*_0_[*M*[η]/100]^1/3^)/*T,*(9)
where *D*_0_ = *kT*/6πη_0_·*R*_h-D_ is the translational diffusion coefficient, *k* is the Boltzmann constant, and *T* is the absolute temperature. The *A*_0_ values for PDMSk-*graft*-PiPrOx are low (Table 1). In particular, they are significantly lower than the theoretical value *A*_0_ = 2.88 × 10^−10^ erg·K^−1^ mol^−1/3^ for the solid sphere. Such low values of the hydrodynamic invariant are characteristic of polymers with a high intramolecular density, such as dendrimers and hyperbranched polymers [55,56,57]. The *A*_0_ values for the macroinitiator agree satisfactorily with the average experimental and theoretical values for flexible chain polymers *A*_0_ = (3.2 ± 0.3) × 10^−10^ erg·K^−1^ mol^−1/3^ [44,53], which indicates that PDMS3 and PDMS10 are in a conformation of the swollen coil.

More rigorous conclusions about the conformation of PDMSk and PDMSk-*graft*-PiPrOx molecules can be drawn by comparing the results obtained with the data for the linear PDMS and poly-2-alkyl-2-oxazolines. Figure 9 on a double logarithmic scale shows the dependences of the intrinsic viscosity [η] and hydrodynamic radii *R*_h-D_ of macromolecules for linear PDMS and poly-2-ethyl-2-oxazoline (PEtOx), obtained by averaging the experimental Mark–Kuhn–Houwink dependences (MKH) for PDMS [58,59,60,61] and PEtOx [50,62] in thermodynamically good solvents. In this case, the MKH dependences were used in the range of the same MM as the samples under study. On the same graphs, the values of the [η] and *R*_h-D_ for PDMSk and PDMSk-*graft*-PiPrOx are presented.

The experimental points corresponding to the values of [η] and *R*_h-D_ for PDMSk lie between the dependences of lg [η] and lg *R*_h-D_ on lg *M* for linear PDMS and PEtOx (Figure 9). These polymers are typical flexible-chain polymers with a Kuhn segment length *A* = 1.5 [63,64] and 1.7 nm [62,65], respectively. Therefore, it can be assumed that PDMSk macroinitiators have an equilibrium rigidity close to that of PDMS and PEtOx. This is consistent with the assumption that the PDMSk are comb-like polymers with a low grafting density of undecyl tosylate side groups.

The intrinsic viscosity of the PDMSk-*graft*-PiPrOx is much lower than the [η] for PDMS and PEtOx at the same MM. A similar trend is observed for the hydrodynamic radii of the studied grafted copolymers. This behavior is typical for polymers with increased intramolecular density, particularly for molecular brushes. For example, the values of [η] and *R*_h-D_ for the grafted copolymers with the main polyimide and side polymethylmethacrylate chains are noticeably lower than the corresponding characteristics for linear polymethylmethacrylate [66].

The conformation of macromolecules of graft copolymers can be determined not only by structural characteristics but also by the thermodynamic quality of the solvent with respect to the backbone and side chains. In selective solvents, to ensure solubility, insoluble fragments must be shielded from the solvent by soluble fragments, which can lead to self-organization at the molecular level and/or aggregation. A similar situation occurs for the investigated copolymers. For them, the solvent used is selective: it is thermodynamically good for side chains, but the PDMS chains dissolve poorly in it. Accordingly, PiPrOx chains tend to shield the branch center from the solvent. It was shown above that for PDMSk-*graft*-PiPrOx copolymers; the *L*_sc_/∆*L* ratio does not exceed 8.4 (Table 2). It is easy to show that for reliable screening of the PDMS chain from the solvent, it is necessary that the ratio *L*_sc_/∆*L* be larger. The increase in *L*_sc_/∆*L* can be caused by the folding of the backbone, which tends to “leave” the solvent inside the volume of the macromolecule. Thus, to ensure the solubility of PDMSk-*graft*-PiPrOx, the main chain is folded, surrounding itself with a dense shell of PiPrOx. Therefore, the PDMSk-*graft*-PiPrOx copolymer molecule in a selective solvent resembles a multi-arm star-shaped polymer molecule with a highly folded backbone serving as the core and PiPrOx acting as arms.

The conformation of the PDMS3-*graft*-PiPrOx and PDMS10-*graft*-PiPrOx macromolecules is very different. Indeed, based on the values of the hydrodynamic radius *R*_h-D_ and the side chain length *L*_sc_ (see Table 1 and Table 2), in the case of PDMS3-*graft*-PiPrOx, the arms have an elongated shape and form a dense shell (Figure 10). In contrast, in the case of PDMS10-*graft*-PiPrOx, the arms are strongly convolved, as indicated by the difference in *R*_h-D_ and *L*_sc_ values, namely *R*_h-D_ << *L*_sc_. Accordingly, the shell of the PiPrOx arms is less dense (Figure 10). In both cases, this shielding is sufficient for the grafted copolymers to dissolve molecularly dispersed in organic solvents. That is, no micelle formation was detected in selective solvents. On the other hand, PDMSk-*graft*-PiPrOx macromolecules resemble unimolecular micelles of the core-shell type.

To quantitatively describe changes in hydrodynamic and conformational characteristics from a linear polymer to a branched one, compression factors are often used. Since there was no scattered light asymmetry even for the highest molar mass weight sample, the values of the radius of gyration and, accordingly, the conformational compression factor
*g =* (*R*_g_)_br_^2^*/*(*R*_g_)_lin_^2^*,*(10)
could not be determined. (Here and below, subscript characters “br” and “lin” mean that the parameter refers to the branched and linear polymer, respectively). Table 3 shows the values of the “viscous” *g** and «hydrodynamic» *h* compression factors:*g** *=* [η]_br_/[η]_lin_(11)
and
*h =* (*R*_h-D_)_br_*/*(*R*_h-D_)_lin_(12)

The values of [η]_lin_ and (*R*_h-D_)_lin_ were obtained by averaging the corresponding parameters for linear PDMS and PEtOx, which were calculated from the MKH equations for these polymers. With an increase in the grafted chain number in PDMSk-*graft*-PiPrOx molecules, the *g** parameter noticeably decreases. This behavior is qualitatively similar to what is predicted theoretically [67,68,69,70] and is observed for polymer stars [69]. Note that the “viscous” compression factor for PDMS10-*graft*-PiPrOx with five grafted chains is close to the *g** values for the star-shaped six-arm PiPrOx [71]. For the “hydrodynamic” contraction factor, there is also a tendency to decrease with increasing the number of grafted PiPrOx chains. However, the difference in the *h* values for PDMS3-*graft*-PiPrOx and PDMS10-*graft*-PiPrOx lies within the experimental error. At the same time, the *h* values for the studied grafted copolymers are lower than this factor for six-arm PiPrOx [71].

### 3.6. Self-Organization of Grafted Copolymers in Aqueous Solutions

Cylindrical brush PDMS10-*graft*-PiPrOx with the largest distance ∆*L* between neighboring grafted PiPrOx chains did not dissolve in water at temperatures from 5 to 65 °C in the concentration range from 0.001 to 0.01 g∙cm^−3^. The denser PDMS3-*graft*-PiPrOx brush exhibited thermoresponsivity in aqueous solutions, as illustrated in Figure 11, which shows the temperature dependence of optical transmission *I**. The temperature *T*_1_ at which the decrease in *I** begins can be considered as the temperature of the onset of phase separation. According to the data obtained by turbidimetry, the phase transition ends at *T*_2_ when *I** becomes minimal.

At low temperatures for aqueous solutions of the grafted copolymer, the distribution of light scattering intensity over hydrodynamic radii was bimodal (Figure 12), i.e., in the PDMS3-*graft*-PiPrOx solutions, there were two types of scattering objects with hydrodynamic radii *R*_f_ (particles responsible for the fast mode) and *R*_s_ (particles responsible for the slow mode). The existence of two or more types of particles in solutions of stimuli-sensitive molecular brushes was observed quite often [32,33,39,72,73,74], and the size and fraction of different particles depended on the grafting density of side chains, their length, and the hydrophilic-hydrophobic balance, as well as on the concentration of the polymer.

No systematic change in the radii *R*_f_ and *R*_s_ was found (Figure 13). The *R*_f_ values, within the experimental error, coincide with the hydrodynamic radii of the macromolecule *R*_h-D_. Thus, it can be concluded that the particles responsible for the fast mode are PDMS3-*graft*-PiPrOx molecules. In addition, it can be assumed that macromolecules in aqueous solutions have the same conformation as in the organic selective solvents; they resemble star-shaped polymer molecules.

The slow mode describes the diffusion of aggregates. At low temperatures, the formation of aggregates is mainly due to the interaction of the main PDMS chains. From general considerations, it is clear that macromolecules are involved in these interactions, in which the grafting density *z* and the length *L*_sc_ of side chains are noticeably lower than the average values of these parameters for the ensemble of macromolecules. Another reason for aggregation may be the interaction of dehydrated 2-isopropyl-2-oxazoline units since the dehydration of PiPrOx can begin at 20 °C [75].

AFM studies confirm the composition of scattering objects that exist in aqueous solutions of PDMS3-*graft*-PiPrOx at low temperatures (Figure 14). In the case of sample preparation by drying an aqueous solution on a mica surface, grain morphology was also observed. The average size of grains was slightly higher in comparison with the sample prepared from organic solvent (30–35 nm). Additionally, AFM images of the sample PDMS3-*graft*-PiPrOx prepared from H_2_O demonstrated some bigger and separately staying spheres with a size of about 130 nm.

At room temperature, the contribution *S*_s_ of aggregates to the integral value of the light scattering intensity for all solutions significantly exceeded the corresponding characteristic *S*_f_ for the macromolecules (Figure 15). At the same time, no dependence of *S*_f_ and *S*_s_ on concentration was found: the average values of *S*_f_ and *S*_s_ were 81 and 19%, respectively. The contribution *I*_i_ of the *i*-th set of particles to the total intensity of light scattering intensity depends on their weight fraction in the solution, their size, and shape [76,77]:*I_i_* ~ *c_i_*∙*R_i_*^x^,(13)
where *c*_i_ and *R*_i_ are the weight concentration and particle radius. The value of the exponent *x* depends primarily on the shape of the particles. For a rough estimate of the fraction of each type of particle in the PDMS10-*graft*-PiPrOx solutions, one can use the sphere model for macromolecules (*x* = 3) and coil (*x* = 2) for aggregates. Calculations show that the relative weight fraction *c**_s_ = *c*_s_/(*c*_s_ + *c*_f_) of aggregates is about 65%, and, accordingly, the relative concentration of macromolecules *c**_f_ = *c*_f_/(*c*_s_ + *c*_f_) ≈ 35% (*c*_f_ and *c*_s_ are weight concentration of particles responsible for the fast and slow modes).

As for other poly-2-alkyl-2-oxazolines, the thermoresponsivity of the studied polymers in aqueous solution is due to an increase in the average degree of dehydration of amide groups on heating and the formation of intra- and intermolecular hydrogen bonds [1,13]. In addition to a change in optical transmission, this is manifested in a change in the light scattering intensity *I* (Figure 16). The *I* value slowly increases with the heating of aqueous solutions of graft copolymers, while in the interval of phase separation, the rate of change of *I* sharply increases. Around the temperature *T*_2_, the light scattering intensity reaches its maximum value.

The dependence of *I* on *T* is explained by the change in the set of scattering objects and their radii on heating (Figure 17). For the PDMS3-*graft*-PiPrOx solutions, the dehydration of poly-2-isopropyl-2-oxazoline leads to the realization of intramolecular hydrogen bonds and the compaction of macromolecules, as evidenced by the decrease in the radii *R*_f_ and *R*_s_ with temperature. Note that the relative change in the radii of molecules *R*_f_ and aggregates *R*_s_ was different. This discrepancy can be explained by the different densities of aggregates and macromolecules, i.e., the different fractions of the polymeric substance per unit volume of the scattering species. For steric reasons, macromolecules are less compact than aggregates.

A decrease in the size of the scattering objects should lead to a decrease in the light scattering intensity; however, as was mentioned above, at *T* < *T*_1_ at all concentrations under study, *I* increased upon heating. This behavior is caused by a change in the weight concentrations of aggregates and macromolecules in solutions, namely, in the considered temperature range *c*_s_* due to a decrease in the fraction *c*_f_* of molecules. Accordingly, on heating, the contribution of aggregates to the integrated light scattering intensity increases, increasing the latter. Thus, at *T* < *T*_1_, two processes occur in aqueous solutions of PDMS3-*graft*-PiPrOx: both the aggregation of molecules and their compaction.

Above the temperature *T*_1_, macromolecules and aggregates with radius *R*_s_ are not detected by the dynamic light scattering, and a sharp increase in the light scattering intensity is due to the appearance of a new type of particles, giant aggregates, which radii are much larger than the size of slow mode aggregates (Figure 17). The *R*_ga_ value increases rapidly, reaching hundreds of nanometers at *T* → *T*_2_. Thus, in the interval *T*_1_ < *T* < *T*_2_, particles that were in solutions at low temperatures combined into new “giant” particles. Aggregation processes are caused by an increase in the dehydration degree of PiPrOx chains upon heating, that is, an increase in the hydrophobicity of PDMS3-*graft*-PiPrOx molecules with temperature.

For most of the studied solutions near the temperature *T*_2_, the *I* values and the aggregate sizes *R*_ga_ reached maximum values. Above *T*_2_, the light scattering intensity and the aggregate radii began to decrease (Figure 16 and Figure 17) due to the compaction of molecules and, accordingly, aggregates. Consequently, at *T* > *T*_2_, compaction prevailed, but a quantitative interpretation of the results under these conditions is impossible since the solutions are turbid and the light scattering method is nonclassical.

Figure 18 shows the concentration dependences of the temperatures of onset of phase separation *T*_1_ for solutions of the copolymer PDMS3-*graft*-PiPrOx. It seems important to study not only the behavior of an individual thermoresponsive polymer as a function of temperature and concentration but also to analyze the thermoresponsivity of a polymer class depending on the architecture parameters of macromolecules. Therefore, Figure 18 also shows the dependences of *T*_1_ on concentration for the PDMS4-*graft*-PiPrOx copolymers [43]. Samples of PDMS4-*graft*-PiPrOx were obtained using the PDMS4 macroinitiator, in which functional undecyl tosylate groups are situated in every fourth PDMS unit. The relative grafting density for PDMS4-*graft*-PiPrOx samples *z* = 0.6 (Table 4); accordingly, the distance between two neighboring side chains in these copolymers is 1.5 times greater than Δ*L* in PDMS3-*graft*-PiPrOx molecules. The *L*_sc_/∆*L* ratios for PDMS3-*graft*-PiPrOx and sample PDMS4-*graft*-PiPrOx-2 differ insignificantly, while for PDMS4-*graft*-PiPrOx-1 with the shortest side chains, the *L*_sc_/∆*L* values are approximately two times lower. For this sample, the mass fraction of hydrophobic fragments ω also differs greatly from ω for the other two PDMSk-*graft*-PiPrOx samples. Note that for the compared samples, the number of side chains *f*_sc_ is large and differs insignificantly.

It is clearly seen that the temperatures, *T*_1_, increased with decreasing concentration. It is typical for thermosensitive polymers to dilute solutions. The rate of change of *T*_1_ with concentration is maximum in the region of strong dilution. In the region of high concentrations, the dependences of *T*_1_ on *c* flatten out, which makes it possible to reasonably reliably estimate the lower critical solution temperature.

Considering the behavior of the concentration dependences of *T*_1_, it can be expected that for the studied PDMS3-*graft*-PiPrOx, the LCST is somewhat lower than 40 °C. This exceeds the LCST of aqueous solutions of the PDMS4-*graft*-PiPrOx sample with a similar *M* [43] by more than 10 °C. The reasons for the difference in LCST for the compared polymers are differences in hydrophobicity ω and relative side chain grafting density *L*_sc_/∆*L* (Table 4). As is known, an increase in ω usually leads to a decrease in the phase separation temperature of aqueous solutions of thermoresponsive polymers. On the contrary, an increase in the *L*_sc_/∆*L* ratio contributes to an increase in the density of the PiPrOx shell, which shields the main PDMS chains from water molecules, and, accordingly, an increase in the LCST.

The LCST for solutions of PDMS3-*graft*-PiPrOx and PDMS4-*graft*-PiPrOx-2 differ slightly (Table 4). A slight decrease in the LCST with the passage from PDMS3-*graft*-PiPrOx to PDMS4-*graft*-PiPrOx-2 can be explained by an increase in *M*, since the remaining characteristics for the compared samples, namely *L*_sc_/∆*L*, ω, and the number of grafted chains *f*_sc_, are similar.

It is interesting to compare the phase separation temperature and LCST values for PDMSk-*graft*-PiPrOx graft copolymers with literature data for thermoresponsive linear and star-shaped PiPrOx. At a given concentration, the *T*_1_ values for aqueous solutions of PDMS3-*graft*-PiPrOx and PDMS4-*graft*-PiPrOx usually do not differ much from *T*_1_ and LCST for linear and star-shaped PiPrOx [78,79], including silicon-containing copolymers [80]. Note that the molar mass of PDMSk-*graft*-PiPrOx is an order of magnitude or higher than the usual MM of linear and star-shaped polymers. Thus, the *M* of the polymer is not the decisive factor determining the values of the phase separation temperatures and LCST. Probably, comparing the *T*_1_ and LCST of thermoresponsive polymers, more factors should be taken into account, for example, the hydrophilic–hydrophobic balance, the distribution of hydrophobic groups over the macromolecule volume, etc.

## 4. Conclusions

Regular and irregular polydimethylsiloxanes have been successfully synthesized. In a regular sample, functional undecyl tosylate groups are located in every third –SiO– unit, and in an irregular one, on average, in every tenth. Using these macroinitiators, molecular brushes with poly-2-isopropyl-2-oxazoline side chains were prepared, where the relative grafting density was 0.7 and 0.6 for the regular and irregular grafted copolymers, respectively.

Using the methods of molecular hydrodynamics and optics, it was shown that the grafting density has almost no effect on the hydrodynamic and conformational behavior of the studied PDMSk-*graft*-PiPrOx samples in dilute solutions since they can be attributed to the class of loose molecular brushes. The shape of PDMSk-*graft*-PiPrOx molecules in selective solvent resembles a star-shaped molecule, the core of which is a collapsed PDMSk backbone, and the PiPrOx chains shield the core from the solvent.

Decreasing the grafting density of the side chain causes the cylindrical brush PDMS10-*graft*-PiPrOx with the largest distance between neighboring grafted PiPrOx chains not to dissolve in water, while the denser brush PDMS3-*graft*-PiPrOx exhibits thermoresponsive behavior in aqueous solutions. The reason for the insolubility of PDMS10-graft-PiPrOx is the hydrophobic interactions of the PDMSk backbone. In the case of PDMS3-*graft*-PiPrOx, these interactions lead to aggregations, and at low temperatures in aqueous solutions of this sample, in addition to isolated macromolecules, there are large loose aggregates. At heating below the phase separation temperature, the hydrodynamic radii of macromolecules and aggregates decrease in PDMS3-*graft*-PiPrOx solutions and the weight concentration of aggregates increases. Therefore, at *T* < *T*_1_, both aggregation and compaction take place. Above the temperature *T*_1_, aggregation becomes the dominant process.

Comparisons of the results obtained for the PDMS3-*graft*-PiPrOx with the literature data for thermoresponsive linear PiPrOx, star-shaped PiPrOx, and grafted copolymers with PiPrOx chains allow us to conclude that the character of self-organization and the value of the LCST depend on numerous factors. For example, in the case of the studied PDMSk-*graft*-PiPrOx, the molar fraction of the hydrophobic fragment and the intramolecular density are important factors in addition to the relative side chain grafting density, more precisely, the ratio of the side chain length to the average distance between grafting points. On the other hand, in the case of poly-2-isopropyl-2oxazolines of complex architecture, the molar mass, determining the LCST for each individual polymer, is not a decisive factor when comparing the thermoresponsivity of PiPrOx with different architectures.

## Figures and Tables

**Figure 1 polymers-14-05118-f001:**
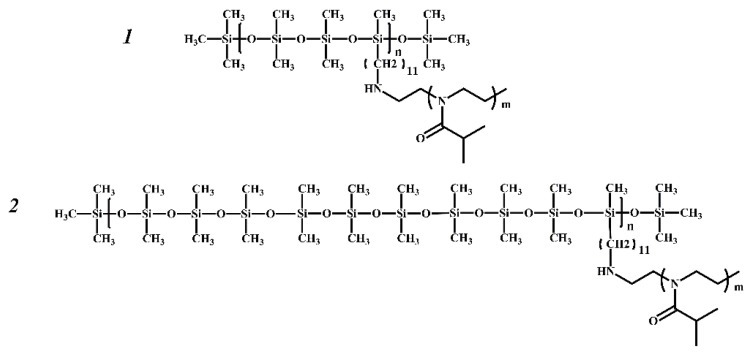
Structural formulas of graft copolymers PDMS3-graft-PiPrOx (**1**) and PDMS10-graft-PiPrOx (**2**).

**Figure 2 polymers-14-05118-f002:**
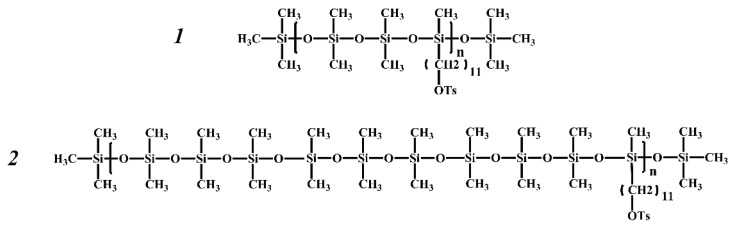
Structural formulas of polydimethylsiloxane macroinitiators PDMS3 (**1**) and PDMS10 (**2**).

**Figure 3 polymers-14-05118-f003:**
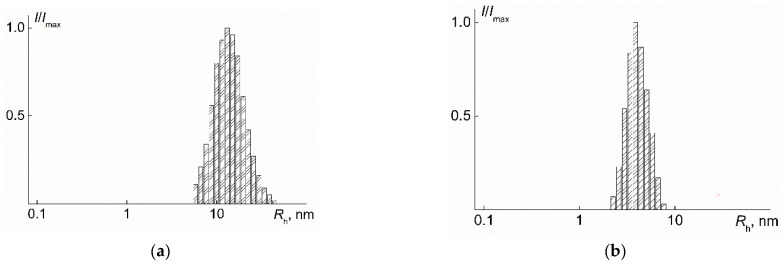
The dependences of the relative intensity *I*/*I*_max_ of scattered light on the hydrodynamic radius *R*_h_ of scattering objects for PDMS3-*graft*-PiPrOx (**a**) at *c* = 0.020 g·cm^−3^ and PDMS10-*graft*-PiPrOx (**b**) at *c* = 0.025 g·cm^−3^ in 2-nitropropane. *I*_max_ is the maximum value of the light scattering intensity at a given *c*.

**Figure 4 polymers-14-05118-f004:**
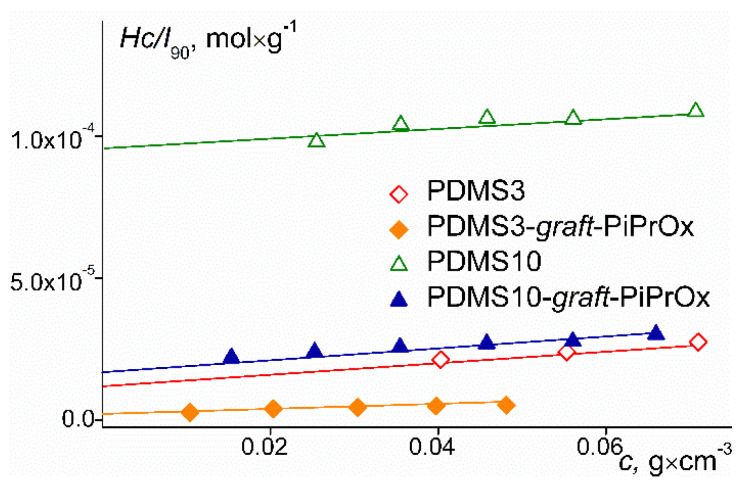
Concentration dependences of inverse excess light scattering intensity *Hc/I*_90_ for investigated samples.

**Figure 5 polymers-14-05118-f005:**
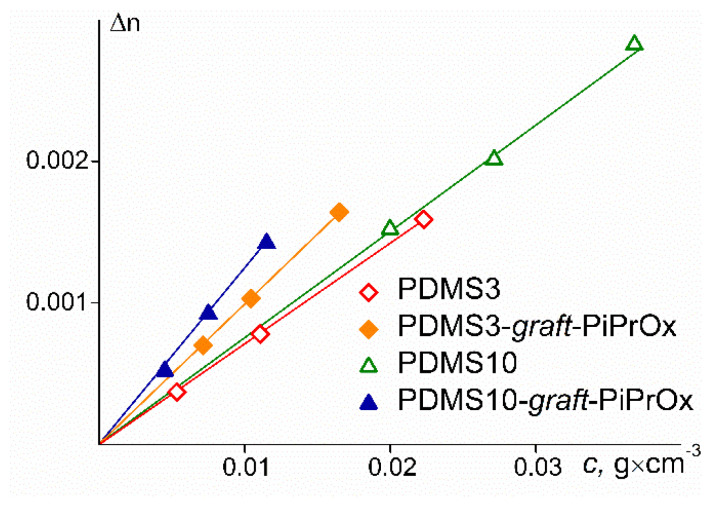
Concentration dependence of *n* and *n*_0_ differences for investigated samples.

**Figure 6 polymers-14-05118-f006:**
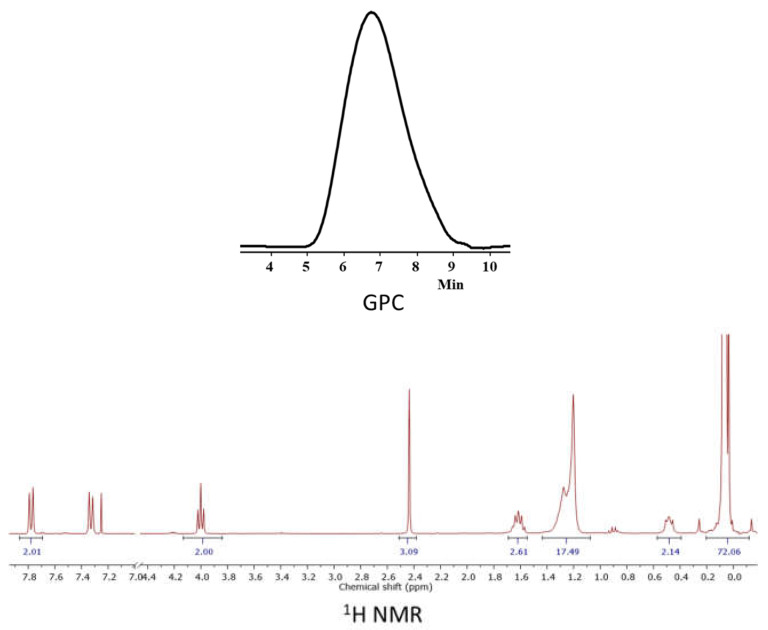
GPC curves and ^1^H NMR spectrum of PDMS10.

**Figure 7 polymers-14-05118-f007:**
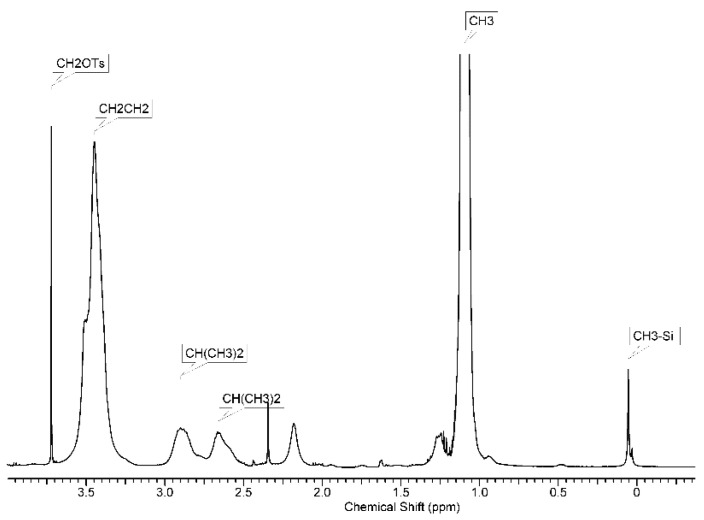
^1^H NMR spectrum of P PDMS3-*graft*-PiPrOx.

**Figure 8 polymers-14-05118-f008:**
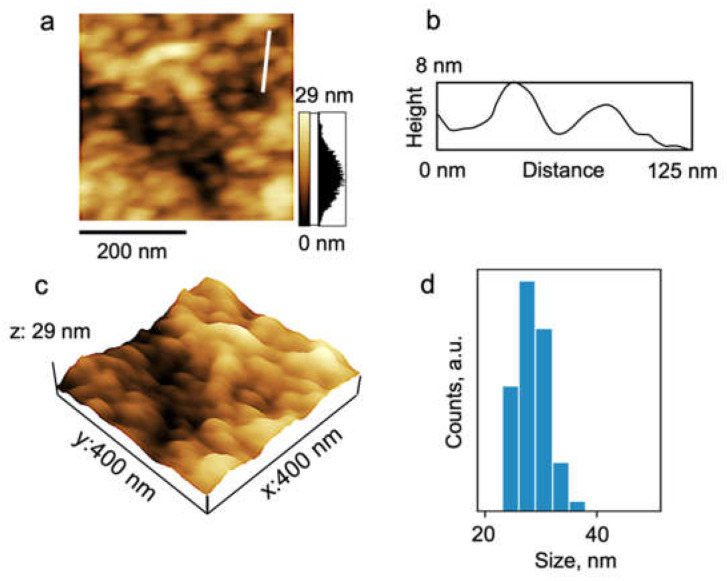
The topography 2D (**a**) and 3D (**c**) images and the height profile (**b**) correspond to the white line traced on the 2D topography image for the PDMS3-*graft*-PiPrOx sample prepared from 2-nitropropane. (**d**) The particle size distribution histogram.

**Figure 9 polymers-14-05118-f009:**
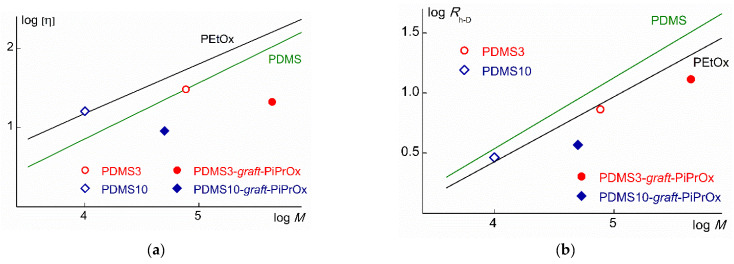
Values of [η] (**a**) and *R*_h-D_ (**b**) as functions on MM for PDMSk and PDMSk-*graft*-PiPrOx. Straight lines are the MKH dependences for the linear PDMS and PEtOx.

**Figure 10 polymers-14-05118-f010:**
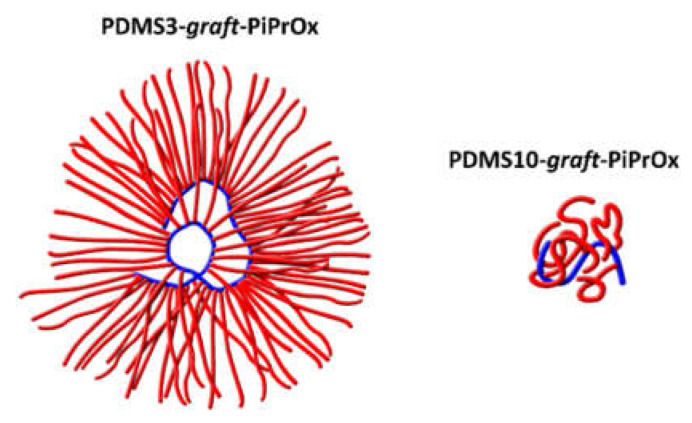
Schematic presentation of the conformation of the studied molecular brushes in 2-nitropropane.

**Figure 11 polymers-14-05118-f011:**
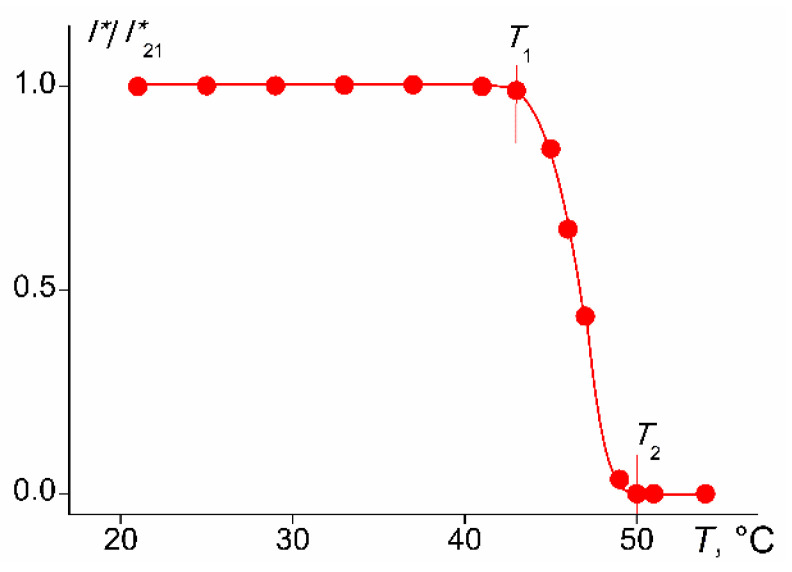
Temperature dependences of relative values of optical transmission *I**/*I**_21_ for the PDMS3-*graft*-PiPrOx solutions at *c* = 0.0010 g∙cm^−3^. *I**_21_ is the l optical transmission at 21 °C.

**Figure 12 polymers-14-05118-f012:**
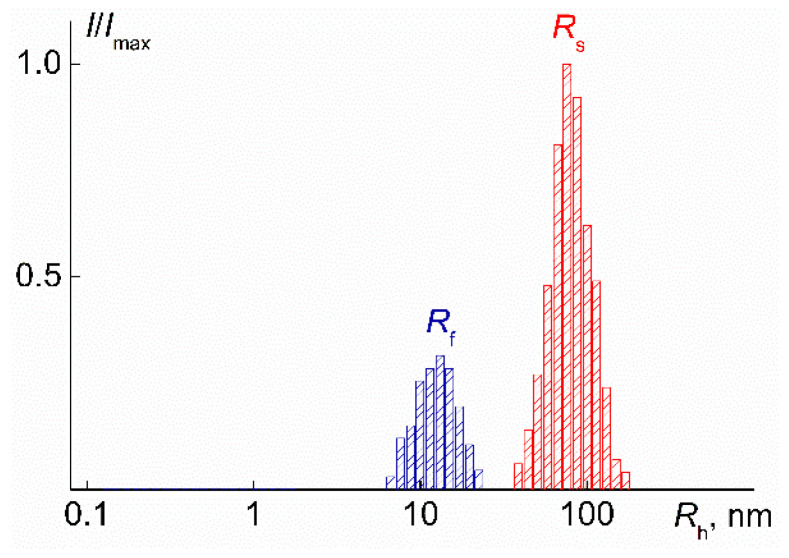
Distribution of light scattering intensity *I*/*I*_max_ over hydrodynamic radii *R*_h_ of the fast and slow modes for PDMS3-*graft*-PiPrOx solutions at *c* = 0.0020 g∙cm^−3^ and *T* = 21 °C. *I*_max_ is the maximum value of the light scattering intensity at a given concentration.

**Figure 13 polymers-14-05118-f013:**
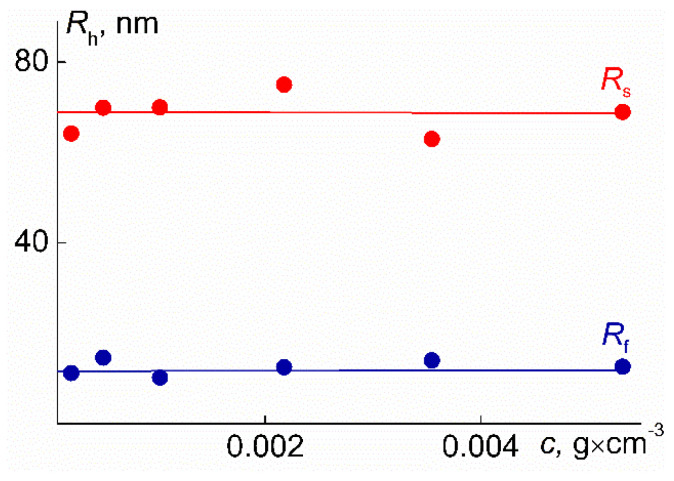
Concentration dependences of the hydrodynamic radii *R*_f_ and *R*_s_ for aqueous solutions of PDMS3-*graft*-PiPrOx solutions at 21 °C.

**Figure 14 polymers-14-05118-f014:**
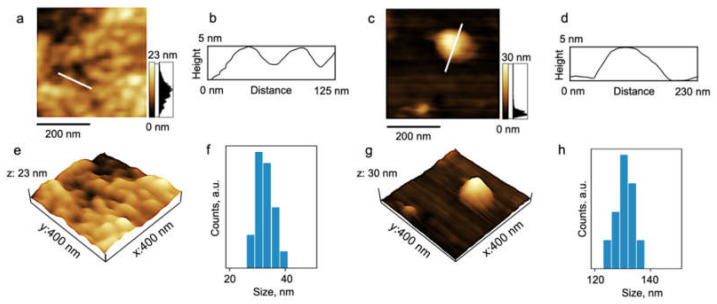
The topography 2D (**a**,**c**) and 3D (**e**,**g**) images and the height profiles (**b**,**d**) correspond to the white lines traced on the 2D topography images for the PDMS3-*graft*-PiPrOx sample prepared from water. (**f**,**h**)–the particle size distribution histograms.

**Figure 15 polymers-14-05118-f015:**
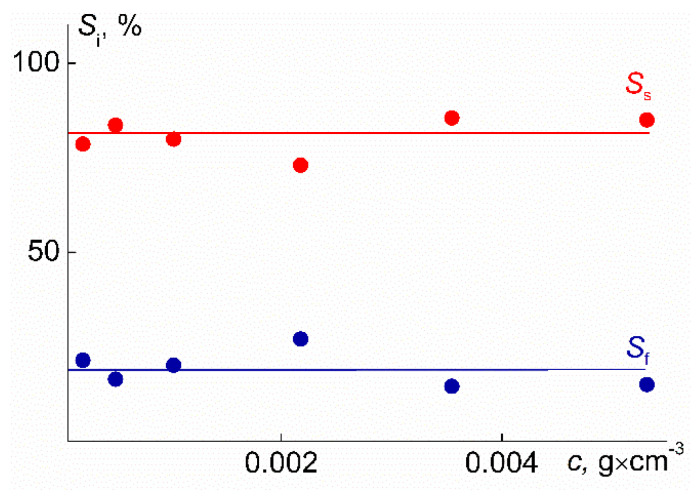
Concentration dependence of the contributions to the light scattering intensity *S*_f_ and *S*_s_ for a PDMS3-*graft*-PiPrOx solutions at 21 °C.

**Figure 16 polymers-14-05118-f016:**
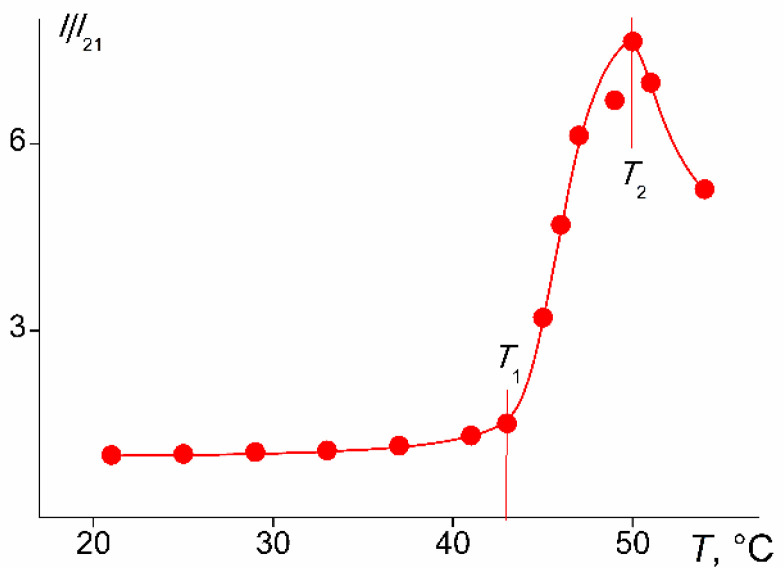
Temperature dependences of the light scattering intensity *I*/*I*_21_ for PDMS3-*graft*-PiPrOx solutions at *c* = 0.0010 g∙cm^−3^. *I*_21_ is the value of scattered light intensity at 21 °C.

**Figure 17 polymers-14-05118-f017:**
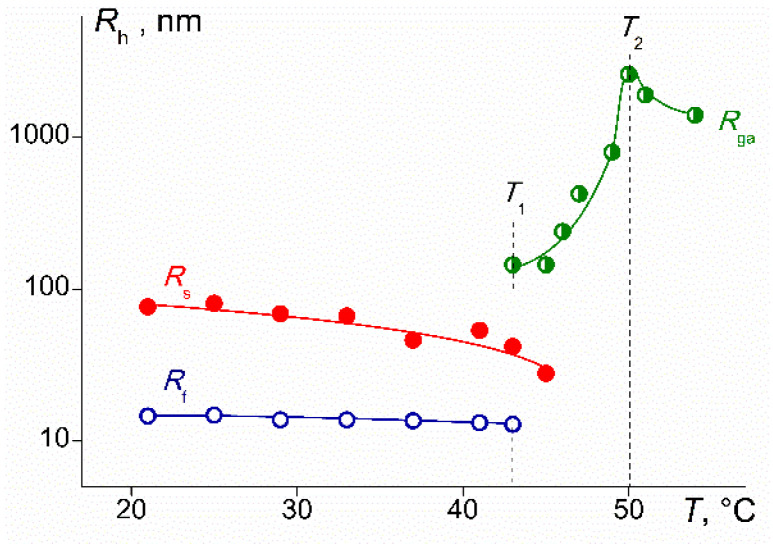
Temperature dependence of the hydrodynamic radii of the fast *R*_f_, slow *R*_s_, and very large *R*_ga_ modes (giant aggregates) for PDMS3-*graft*-PiPrOx solutions with a concentration *c* = 0.0010 g∙cm^−3^.

**Figure 18 polymers-14-05118-f018:**
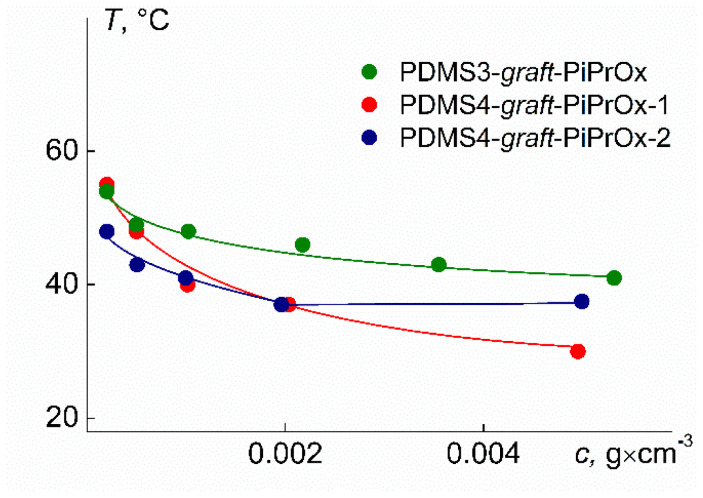
Concentration dependences of the temperature *T*_1_ for aqueous solutions of PDMS3-*graft*-PiPrOx and PDMS4-*graft*-PiPrOx. Data from [43].

**Table 1 polymers-14-05118-t001:** Molar mass and hydrodynamic characteristics of the macroinitiators and grafted copolymers.

Title 1	*M*_w_, g∙mol^−1^	*R*_h-D_, nm	*dn*/*dc*, cm^3^∙g^−1^	$\bar{v}$, cm^3^∙g^−1^	[η], cm^3^∙g^−1^	*A*_0_∙10^−10^, Erg∙K^−1^∙mol^−1/3^
PDMS3	77,000	6.3	0.0703	0.968	24	3.1
PDMS3-*graft*-PiPrOx	440,000	13.0	0.1156	0.935	21	1.5
PDMS10	10,000	2.9	0.0756	0.979	16	3.0
PDMS10-*graft*-PiPrOx	50,000	3.7	0.1256	0.929	9.0	2.5

**Table 2 polymers-14-05118-t002:** Structural characteristics of the macroinitiators and grafted copolymers.

Title 1	λ_bb_, nm	*z*	Δ*L*, nm	*L*_sc_, nm	*L*_sc_/∆*L*	*f* _sc_	ω, Mass%
PDMS3	0.78		-	2.1	-	-	-
PDMS3-*graft*-PiPrOx	0.78	0.7	1.1	9.3	8.4	76	17.5
PDMS10	2.6		-	2.1	-	-	-
PDMS10-*graft*-PiPrOx	2.6	0.6	4.3	14.2	3.3	5	20.0

**Table 3 polymers-14-05118-t003:** Structural characteristics of the macroinitiators and grafted copolymers.

Polymer	*g**		*h*	
	PDMS	PEtOx	PDMS	PEtOx
PDMS3-*graft*-PiPrOx	0.20	0.13	0.63	0.44
PDMS10-*graft*-PiPrOx	0.41	0.23	0.58	0.42

**Table 4 polymers-14-05118-t004:** Structural characteristics of macroinitiators and grafted copolymers.

Title 1	*z*	*M*_w_, g·mol^−1^	Δ*L*, nm	*L*_sc_, nm	*L*_sc_/∆*L*	*f* _sc_	ω, Mass%	LCST, °C
PDMS3-*graft*-PiPrOx	0.7	440,000	1.11	9.3	8.4	76	17.5	<40
PDMS4-*graft*-PiPrOx-1 *	0.6	400,000	1.73	6.7	3.9	86	28.8	≤27
PDMS4-*graft*-PiPrOx-2 *	0.6	700,000	1.73	12.7	7.4	86	16.2	≤37

* Values of characteristics are given according to data obtained in other article. Data from [43].

## Data Availability

Not applicable.

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
