# Peer review of "Amphiphilic Molecular Brushes with Regular Polydimethylsiloxane Backbone and Poly-2-isopropyl-2-oxazoline Side Chains. 3. Influence of Grafting Density on Behavior in Organic and Aqueous Solutions"

_polymers, 2022, doi:10.3390/polym14235118_

Round 1

Reviewer 1 Report

Dear Authors

The manuscript has good novelty and it could be published after considering following major revision:

1-     More AFM height and phase image must be included in the paper. 2D AFM height and phase images.

2-     Procedure of calculation of grafting density must be shown exactly on the HNMR profile.

3-     There are numerous grammatical and spelling errors that must be corrected.

4-     Moderate similarity (38%) must be corrected before publishing.

Sincerely

Author Response

Reliever 1

Dear Authors

The manuscript has good novelty and it could be published after considering following major revision:

Dear Reviewer,

Thank you very much for your comments on our manuscript. We carefully read them and revised the manuscript, making the necessary changes.

(1)    More AFM height and phase image must be included in the paper. 2D AFM height and phase images.

AFM study was carried out in the dynamic mode of microscope operation. This mode does not imply simultaneous recording of the phase channel along with topography channel. Moreover, polymer particles can be easily distinguished on the mica surface as a hills with sizes confirmed with other methods. Thus, additional phase registration does not need. For more clear presentation AFM results we included additional 2D AFM height images and particle size distribution histograms in Figures 8 and 14 of the revised manuscript.

(2)    Procedure of calculation of grafting density must be shown exactly on the HNMR profile.

We added an NMR plot and corrected the manuscript.

(3)    There are numerous grammatical and spelling errors that must be corrected.

In view of your comments, we have checked and corrected the manuscript.

(4)    Moderate similarity (38%) must be corrected before publishing.

We have corrected the manuscript, but it should be kept in mind that the main amount of repetition with previous articles is contained in the experimental part. It is not possible to make major changes or shorten this experimental part, as the reader would be uncomfortable with the manuscript due to the lack of visibility of the experiment.

Summary:

We answered all comments and made necessary changes in manuscript. All necessary corrections and additions in the manuscript are marked in yellow for easy reference.

Reviewer 2 Report

The paper submitted by Rodchenko et al. investigates the synthesis of two amphiphilic graft copolymer samples having a brush molecular conformation. The backbone is based on hydrophobic PDMS whereas the side chains are hydrophilic poly(2-isopropyl oxazoline).

The manuscript is clear, well written and the conclusions are supported by the results. However, some corrections are needed in order to increase the overall quality of the paper:

1. the introduction section must be completed with several new references, such as: https://doi.org/10.1016/j.progpolymsci.2017.06.001; https://doi.org/10.3390/polym13091351; https://doi.org/10.1016/j.eurpolymj.2014.01.029; https://doi.org/10.1021/acs.macromol.9b01662; https://doi.org/10.3390/polym10010062

2. line 73: amphiphilic copolymers

3. it will be very useful if the authors could provide a schematic representation of the micelles in both organic and aqueous media in order to better visualize the difference between the two copolymer samples.

4. this type of graft copolymer are well known to form unimolecular micelles. however, the authors have not taken into consideration this possibility. why they have excluded this possibility especially when the Mw of the copolymers is quite high...

Author Response

Reliever 2

Comments and Suggestions for Authors

The paper submitted by Rodchenko et al. investigates the synthesis of two amphiphilic graft copolymer samples having a brush molecular conformation. The backbone is based on hydrophobic PDMS whereas the side chains are hydrophilic poly(2-isopropyl oxazoline).

The manuscript is clear, well written and the conclusions are supported by the results. However, some corrections are needed in order to increase the overall quality of the paper:

Dear Reviewer,

Thank you very much for your comments on our manuscript. We carefully read them and revised the manuscript, making the necessary changes.

(1)    the introduction section must be completed with several new references, such as: https://doi.org/10.1016/j.progpolymsci.2017.06.001; https://doi.org/10.3390/polym13091351; https://doi.org/10.1016/j.eurpolymj.2014.01.029; https://doi.org/10.1021/acs.macromol.9b01662; https://doi.org/10.3390/polym10010062.

Thank you for your comment. We have reviewed the articles you suggested, and we have used some of them in our literature review.

(2)    line 73: amphiphilic copolymers.

Thank you, we corrected the manuscript.

(3)    it will be very useful if the authors could provide a schematic representation of the micelles in both organic and aqueous media in order to better visualize the difference between the two copolymer samples.

No micelle formation was detected in selective organic solvents. On the other hand, macromolecules resemble star-like micelles in shape, i.e., they can be considered as unimolecular micelles of the core-shell type.

Based on the values of the hydrodynamic radius and side chain length in the case of PDMS3-graft-PiPrOx arms are straightened and form a dense shell. In contrast, in the case of PDMS10-graft-PiPrOx, the arms are curved and the surrounding shell is less dense. But in both cases this shielding is sufficient for the grafted copolymers to dissolve in organic solvents.

When moving to aqueous solutions, such a dense shell, as in the case of PDMS3-graft-PiPrOx, is enough for individual macromolecules to be present in the solution. The small shell density in the case of PDMS10-graft-PiPrOx leads to the fact that in aqueous solutions polymer-polymer interactions occur immediately and the graft copolymer does not dissolve in water. These factors indicate that water is a worse solvent than 2-nitropropane for investigated PDMSk-graft-PiPrOx.

(4)         this type of graft copolymer are well known to form unimolecular micelles. however, the authors have not taken into consideration this possibility. why they have excluded this possibility especially when the Mw of the copolymers is quite high...

Thank you very much, we have made changes to the manuscript regarding aggregation, in particular describing unimolecular micelles in organic solvents. See also answer 3. question 3 and 4 overlap, for which we thank the reviewer for sharpening our attention. Therefore, we partially use the answer to the previous question.

Summary:

We answered all comments and made necessary changes in manuscript. All necessary corrections and additions in the manuscript are marked in yellow for easy reference.

Round 2

Reviewer 1 Report

Dear Authors

The manuscript could be published.

Sincerely